# A Retrospective Study of Risk Factors, Mortality, and Treatment Outcomes for Infections with Carbapenemase-Producing *Enterobacterales* in a Tertiary Hospital in Havana, Cuba

**DOI:** 10.3390/antibiotics11070942

**Published:** 2022-07-14

**Authors:** Haiyang Yu, Alberto Hernández González, Gonzalo Estévez Torres, María Karla González Molina, Marcia Hart Casares, Xu Han, Waldemar Baldoquín Rodríguez, Dianelys Quiñones Pérez

**Affiliations:** 1Pedro Kourí Institute of Tropical Medicine, Havana 11400, Cuba; yuheroyoung@hotmail.com (H.Y.); marikarla@ipk.sld.cu (M.K.G.M.); aselahan@hotmail.com (X.H.); wbaldoquin@infomed.sld.cu (W.B.R.); 2Hermanos Ameijeiras Hospital, Havana 10400, Cuba; dralbertohernandez4@gmail.com (A.H.G.); epi@hha.sld.cu (G.E.T.); mlhart63@yahoo.com (M.H.C.)

**Keywords:** carbapenemase, *Enterobacterales*, Cuba, NDM, risk factor, mortality, treatment

## Abstract

(1) Background: The spread of carbapenem-resistant *Enterobacterales* in hospitals constitutes an important epidemiological and therapeutic problem that especially affects vulnerable patients such as perioperative patients. (2) Methods: We conducted a descriptive, observational, retrospective case-control study of patients infected with carbapenemase-producing carbapenem-resistant Enterobacterales (CP-CRE) and carbapenem-susceptible Enterobacterales during the perioperative period in a tertiary hospital. (3) Results: Metallo-β-lactamase was detected in all 124 CRE isolates, with NDM-type carbapenemase being dominant, while 3 isolates coproduced KPC-type enzyme and showed high resistance rates against all antibiotics except colistin (25.2%). By analyzing the risk factors for infection, steroid use (OR: 3.22, *p* < 0.01), prior use of two or more antibiotics (OR: 4.04, *p* = 0.01), prior use of broad-spectrum cephalosporins (OR: 2.40, *p* = 0.04), and prior use of carbapenem (OR: 4.77, *p* = 0.03) were found to be independent risk factors for CP-CRE infection. In addition, in this study, we observed that the clinical outcomes of bloodstream infections and pneumonia associated with CP-CRE posed higher mortality risks. However, by analyzing the associations between treatment options and mortality, it was found that, in bloodstream infections caused by CP-CRE, colistin-based regimens showed a significant advantage (PR = 0.40, *p* = 0.03). (4) Conclusions: High mortality is associated with nosocomial infections in the perioperative period caused by carbapenemase-producing Enterobacterales, the dissemination of which in health care settings in Cuba remains a public health challenge.

## 1. Introduction

In recent years, carbapenem-resistant *Enterobacterales* (CRE) has become increasingly prevalent as an etiologic agent of healthcare-associated infections (HAIs) and presents a major clinical impact due to the very limited therapeutic options available [1].

The emergence of CRE may be caused by a combination of mechanisms: the production of carbapenemase enzymes, the alteration of membrane permeability, and active expulsion systems. However, enzymatic production is their main mechanism of resistance [2]. Several different carbapenemases can be encountered among carbapenemase-producing carbapenem-resistant *Enterobacterales* (CP-CRE), including class A serine β-lactamases (e.g., KPC-type enzymes), [3] class D serine β-lactamases (e.g., OXA-48 and related enzymes), and class B metallo-β-lactamases (e.g., NDM-, VIM-, and IMP-type enzymes) [4].

According to the worldwide surveillance of *Enterobacterales* by SENTRY, CRE infections have shown a significant increase globally, with a greater impact in Latin America (rates increasing from 0.8% in 1997 to 6.4% in 2016). The most predominant infections with CRE among hospitalized patients are pneumonia and bloodstream infection (BSI) (3.3% and 2.5%, respectively) while the prevalence of skin and soft tissue infection (SSTI) and urinary tract infection (UTI) is 1.8% and 1.2%, respectively [5].

Therefore, the prevention and control of CRE-associated infection is an important research topic. Currently, although general risk factors associated with the development of CP-CRE infection have been described, the most closely associated factors are personal history of disease, invasive interventions, and antibiotic use [6,7]. However, there are few studies relevant to perioperative patients (the entire period from admission to hospital, before and after surgery, and up to discharge from hospital).

The present study targeted nosocomial infections with CRE in perioperative patients in a tertiary hospital in Cuba. We analyzed the microbiological characteristics of CP-CRE and explored the clinical-epidemiological characteristics of CP-CRE infection episodes.

## 2. Results

### 2.1. Microbiological Study

The confirmation of CP-CRE and the characterization of carbapenemases are shown in Table 1. A wide dissemination of carbapenemases was observed in the *Enterobacterales* family, with the highest prevalence of isolates corresponding to *K. pneumoniae*, followed by other species such as *E. cloacae*, *E. coli*, *K. aerogenes*, and *S. marcescens*. The identification of the type of carbapenemase involved was performed using two methods. A total of 124 metallo-β-lactamase-producing isolates were identified. Of these, 55 isolates were processed by PCR, with 52 isolates with a single NDM carbapenemase being detected and 3 isolates with co-production of NDM and KPC found. The remaining isolates (69 isolates) were confirmed as NDM enzyme producers by immunochromatographic testing.

The percentages of resistant isolates are demonstrated in Table 2. All isolates (*n* = 124) were resistant to all β-lactam antibiotics tested, except aztreonam. The rates of resistance to fosfomycin, ciprofloxacin, gentamicin, amikacin, trimethoprim-sulfamethoxazole, and tigecycline were, respectively, 71.8%, 87.1%, 92.8%, 91.9%, 93.5%, and 80.6%. Overall, high resistance rates were observed against all antibiotics except for colistin (CST) at 25.2%, with, respectively, 26.1%, 31.1%, and 25.0% of the *K. pneumoniae*, *E. cloaceae*, and *K. aerogenes* strains displaying colistin MIC ≥4 μg/mL while it was extremely rare among *E. coli* strains.

### 2.2. Population Study

A total of 88 cases of CP-CRE healthcare-associated infections were diagnosed in the study period. The age distribution of cases ranged from 20 to 87 years, with a mean of 55.5 ± 14.8. Regarding the distribution of cases according to hospital services, more than half of the patients were from intensive care units (28.4%) and urology departments (25%). In terms of the infection site, the majority CP-CRE isolates were derived from blood (34.1%), urine (34.1%), and surgical wounds (20.5%). Patients infected with carbapenem-susceptible *Enterobacterales* (CSE) were consecutively selected in a 1:1 ratio for controls (88 controls) and matched for age, services received, and types of infection. The *p*-values of all these variables were greater than 0.05, and there were no statistically significant differences between cases and controls with respect to age, services received, and infection sites (Table 3).

### 2.3. Risk Factors

The risk factors associated with CP-CRE and CSE infections are shown in Table 4. By univariate analysis, CP-CRE infection was found to be associated with the length of hospitalization (*p* < 0.01), prolonged derivation (*p* = 0.03), nasogastric tube (*p* = 0.01), mechanical ventilation (*p* = 0. 03), transfer from another healthcare facility (*p* = 0.02), steroid use (*p* < 0.01), prior use of two or more antibiotics (*p* < 0.01), prior use of third- or fourth-generation cephalosporins (*p* < 0.01), prior use of aminoglycosides (*p* < 0.01), and prior use of carbapenem (*p* < 0.01). In multivariate analysis, prior steroid use (OR: 3.22, 95% CI: 1.36–7.66, *p* < 0.01), prior use of two or more antibiotics (OR: 4.04, 95% CI: 1.40–11.71, *p* = 0.01), prior use of extended-spectrum cephalosporins (OR: 2.40, 95% CI: 1.06–5.44, *p* = 0.04), and prior use of carbapenem (OR: 4.77, 95% CI: 1.17–19.35, *p* = 0.03) were the independent risk factors for the infection of CP-CRE.

### 2.4. All-Cause 60-Day Mortality after CP-CRE Acquisition

The 60-day all-cause mortality in case group patients was 21.6% (19/88), compared to CSE-infected patients who had a mortality of 10.2% (9/88). Regarding the risk of mortality between the two groups, the *p*-value was close to the limit and was still not significant (HR = 2.094, *p* = 0.068) (Figure 1). Among those infected with CP-CRE, when comparing the risk of mortality in the different sites of infection, the highest risk was found in pneumonia, followed by bloodstream infection, with a *p*-value < 0.001 (Figure 2).

### 2.5. Evaluation of Treatment Options for CP-CRE Infections

Regarding the treatment for CP-CRE infection, 39.8% (35/88) of patients were treated with monotherapy while, among the combination regimens, colistin-based (38.6%), tigecycline-based (5.7%), and colistin-tigecycline-based (5.7%) regimens were mainly used. By univariate analysis, a colistin-based regimen was found to be protective against mortality caused by CP-CRE bloodstream infections (PR = 0.40, *p* = 0.03), whereas in pneumonia, tigecycline-based or colistin-tigecycline-based regimens were found to potentially be protective against mortality but not to a statistically significant level (PR = 0.52, *p* = 0.28) (Table 5). No mortality occurred in urinary tract infections and surgical site infections either with monotherapy or with different combination regimens. In a limited number of cases of intra-abdominal infection (two cases) and intracranial infection (one case), patients were treated with colistin-based regimens and there were no deaths.

## 3. Discussion

Carbapenem resistance among *Enterobacterales* is an emergent phenomenon of great importance in clinical and public health terms. According to the report of ReLAVRA (the Latin American Antimicrobial Resistance Surveillance Network), from 2010 to 2019, a systematic increase in resistance was reported in Latin America and the Caribbean, reaching a prevalence of above 60% in some countries [8].

Faced with this progressive public health problem, exploring the risk factors for the prevention and control of these infections, and seeking the best treatment option have become essential research directions.

Previous studies have reported similar risk factors associated with CRE infection associated with ICU admission, central venous catheter use, solid-organ or stem cell transplantation, mechanical ventilation, and exposure to extended-spectrum antibiotics [9,10,11]. In this study, the independent factors found to be associated with CP-CRE infection included steroid use and prior use of two or more antibiotics (with a course of more than seven days). Among antibiotics, prior exposure to cephalosporins (third or fourth generation) or carbapenem (with a course of more than seven days) was associated with CRE-PC infection. However, another case-control study of CRE and CSE found that carbapenem use, length of hospitalization, and invasive procedures were also independent risk factors [12]. In addition, it is important to note that, in this study, because the control group selected was CSE-infected patients, these risk factors may be involved in the conversion of carbapenem-susceptible to carbapenem-resistant *Enterobacterales*, but more rigorous controlled studies are required to demonstrate this.

In the present study, NDM-type carbapenemases were detected in all isolates. Although KPC carbapenemase was first detected in the hospital in 2011, an increase in NDM was detected during 2013 according to local surveillance with 5.9% metallo-β-lactamases compared to 1.2% KPC [13]. The metallo-β-lactamases were confirmed as NDM at the National Reference Laboratory for Healthcare-Associated Infections (NRL-HAIs) of Cuba. However, this is in contrast to a review (2021) where KPC-type-producing *Enterobacterales* were found to be widely disseminated in Latin America and the Caribbean, although NDM-producing *Enterobacterales* have been reported successively in several countries in the region [14]. Worldwide, NDM producers are most common in Asia and Europe [15], but Cuba is also heavily affected by NDM-producing *Enterobacterales* [16,17], possibly due to travel from endemic areas and resulting in silent local dissemination. In addition, since 2019, more reports from the National Reference Laboratories members of ReLAVRA have issued alerts about the increase in the number of isolates expressing double carbapenemases, especially the co-production of NDM and KPC [18,19,20]. Cuba is no exception from this problem; three *Klebsiella* spp. with coproduction of NDM and KPC were detected in the present study.

The NDM-type carbapenemase is encoded by the plasmid gene blaNDM, which is easily transferable between bacteria, and its rapid spread may have major epidemiological repercussions worldwide. In addition, the resistance patterns of *Enterobacterales* with the NDM enzyme present extensive resistance to β-lactam antibiotics and are not inhibited by β-lactamase inhibitors, except for aztreonam. However, especially when co-expression is present, some studies have shown that there are remarkable associations between *bla*NDM genes and multiple other resistance genes, such as genes encoding extended-spectrum β-lactamases (ESBLs), AmpC, or other classes of carbapenemases (KPCs) that may lead to resistance to aztreonam [21]. Additionally, combination with ribosomal rRNA methylases (16S-RMTases) can confer a high level of resistance to all aminoglycosides [22]. Most clinical bacteria carrying the NDM gene are only susceptible in vitro to colistin, tigecycline, and fosfomycin, or to all [23]; this view is generally consistent with the results of the present study. The 124 isolates were found to be resistant to almost all β-lactam antibiotics, with approximately 90% of the strains being resistant to fluoroquinolone, aminoglycosides, and sulfonamide, while the rates of resistance to colistin, fosfomycin, and tigecycline were 25.2%, 71.8%, and 80.6%, respectively. However, it should be noted that, although the rate of colistin resistance in this study was relatively low, it still far exceeded the overall level of colistin resistance in Latin America and even reached that seen in some European countries (Spain, Italy, Greece, etc.) with relatively high levels of colistin resistance. A review study (2022) showed that the incidence of colistin-resistant human clinical Enterobacterales increased between 2014 and 2019 from 2.7 in Latin America to 4.3% while the colistin resistance rates in these European countries have stayed between 20% and 40% [24].

Due to the limited treatment options available, infection with CP-CRE is associated with high mortality rates. The present study documents an overall mortality rate at 60 days after CP-CRE infection of 21.6%. In particular, a higher risk of mortality was found in bloodstream-associated infections and pneumonia. However, other studies report even higher mortality rates, such as a review study (2016) reporting mortality rates between 30% and 75% for CP-CRE infections [25]. In addition, this study compared the probability of survival among patients infected with CP-CRE and CSE. For patients with the same underlying conditions (age, Charlson index, and hospitalization services), the mortality risk of patients infected by CP-CRE was twice as high as that of patients infected by CSE in the hospital setting, although the *p*-value was in the borderline region (IC 95%, *p* = 0.068).

The optimal treatment for CP-CRE infections has not yet been defined, and treatment usually involves the use of tigecycline, colistin, amikacin, and fosfomycin, either alone or in combination with each other and with carbapenems [26]. In this study, the majority of patients with CP-CRE received combination therapy, mainly with a colistin-based, tigecycline-based, or colistin-tigecycline-based regimen. The analysis revealed that different combination regimens may each have advantages in different types of infection. Colistin-based regimens showed a significant advantage (*p* = 0.03) in bloodstream infections caused by CP-CRE, whereas regimens containing tigecycline showed some advantage in pneumonia caused by CP-CRE, although this was not statistically significant (*p* = 0.28). This is related to the tissue concentrations of antibiotics at the target site contributing to therapeutic effects, and some studies have shown that it is not possible to measure colistin in bronchoalveolar lavage after giving repeated IV doses of 2 million international units (MIU) of colistin every 8 h to critically ill patients while tigecycline has a large volume of distribution (>12 L/kg), penetrating well into the lungs and with low concentrations in serum [27,28]. Regarding abdominal and intracranial infections, due to the extremely limited number of cases observed, no deaths were found in the three patients treated with colistin-based regimens, but more data are needed to verify this observation. Meanwhile, in urinary tract infections and surgical site infections, no deaths occurred with the use of either monotherapy or combination therapy. However, the vast majority of studies and different expert consensuses currently recommend the use of combination therapy for patients with CP-CRE infection [29,30]. In a meta-analysis study (2018) that evaluated the effect of treatments on mortality outcomes in patients with severe CP-CRE infections, monotherapy led to an increased risk of mortality, and patients receiving only one antimicrobial had twice the probability of mortality compared to those treated with multiple active antibiotics. This risk increased markedly in patients with bacteremia or generalized sepsis, where those treated with monotherapy were 3.8 times more likely to experience mortality compared to patients receiving combination therapy [31]. On the other hand, combination therapies, in addition to their synergistic antibacterial effects to improve efficacy, can also be effective in preventing the evolution of resistance [32]. For MBL-producing *Enterobacterales*—in particular, those that co-produce serine-type β-lactamases—treatment is more challenging. Currently, studies related to treatment options for CP-CRE usually focus on KPC- or OXA-48 carbapenemase-producing *Enterobacterales*, and it has been found that, in addition to combinations based on tigecycline, polymyxin, or carbapenems, which present a synergistic effect, newer drugs such as ceftazidime-avibactam, meropenem-vaborbactam, and plazomycin possess greater advantages against carbapenemase KPC- or OXA-48-producing *Enterobacterales* [26,30]. Therefore, preclinical and anecdotal clinical data support the use of aztreonam in combination with avibactam (aztreonam plus ceftazidime-avibactam or a new drug combination aztreonam–avibactam) against these pathogens because aztreonam is a monobactam stable to hydrolysis by MBLs and avibactam is a β-lactam inhibitor that effectively inhibits serine carbapenemases. However, other aztreonam-based combinations have not been explored [33,34].

Limitations: This study included a relatively small number of patients; since it was a single-center study, these results may not be generalized to other hospitals where different factors could contribute to similar infections. However, this study was necessary due to the observation of a rising trend of infections due to CP-CRE in Cuba, and it is the first study to address this issue for perioperative patients in this region.

## 4. Materials and Methods

### 4.1. Study Design and Description of the Institution

We conducted a single-center retrospective study with a descriptive and observational analytical case-control method (patients with CP-CRE infection were included as cases and compared with controls who were identified as patients infected with carbapenem-susceptible *Enterobacterales* to explore the risk factors for healthcare-associated infection due to CP-CRE. We further assessed mortality related to the same episode of infections in perioperative patients, and the microbiological characteristics of isolates, during the period 2017–2021.

All patients originated from a clinical-surgical hospital in Havana that provides third-level health care for the country’s National Health System. The institution has 513 beds, including a clinical area (253), surgical area (235), and critical care area (25).

### 4.2. Microbiological Study

A total of 124 CRE isolates from the clinical-surgical hospital were sent to NRL-HAIs of the Pedro Kouri Institute of Tropical Medicine during the period from 2017 to 2021. Species verification was performed by conventional microbiological methods according to the Diagnostic Procedures Operations Manual (MOPD, Havana, Cuba) of the NRL-HAIs. The Kligler and oxidase tests were performed initially, and later, the following culture media were used for species identification: Simmons citrate agar (Biolife, Milan, Italy), Christensen urea agar (Biolife, Milan, Italy), mobility-indole agar (Biolife, Milan, Italy), malonate sodium (Biolife, Milan, Italy), lysine decarboxylase broth (Biolife, Milan, Italy), and ornithine decarboxylase broth (Biolife, Milan, Italy). The Epsilon-test (E-test) method was used on Müeller Hinton agar (Biolife, Milan, Italy) to detect imipenem, meropenem, ampicillin-sulbactam, piperacillin-tazobactam, ceftazidime, ceftriaxone, cefepime, aztreonam, amikacin, gentamicin, ciprofloxacin, trimethoprim-sulfamethoxazole, fosfomycin, and tigecycline for susceptibility. Colistin was evaluated using the disk elution method. The results were interpreted according to the criteria established by the CLSI (2021) and EUCAST (2022). The detection and characterization of the carbapenemase type were conducted using the phenotypic method, immunochromatographic method, and molecular method with polymerase chain reaction (PCR). The phenotypic method used was the combined tablet method for KPC-MBL. We used the Confirm ID Pack (Rosco Diagnostica, Denmark), for which the manufacturer’s instructions were followed. For the immunochromatographic method, the commercial kit RESIST-4. O.K.N.V (Coris BioConcept^®^, Gembloux, Belgium) was used according to the manufacturer’s instructions. PCR was performed to determine the presence of *bla*KPC, *bla*NDM, *bla*IMP, *bla*VIM, *bla*SPM, *bla*GIM, *bla*SIM, and *bla*OXA-48 genes using the protocols and conditions described in Nordmann (2011) [35]. Briefly, 8 pairs of primers were designed to amplify internal fragments with sizes from 232 to 798 bp (Table 6). The thermal cycling settings were 30 cycles at 9 °C for 30 s, 55 °C for 1 min, and 72 °C for 30 s, and one cycle at 72 °C for 5 min. A positive control was used for the amplification of each gene. The PCR products were separated by electrophoresis on 1.5% agarose gel (AppliChem, Darmstadt, Germany) and visualized using UV transillumination (Uvitec, Cambridge, UK).

### 4.3. Description of Populations

After the verification of 124 strains, we initially consolidated the samples based on the information in the corresponding medical records. We determined the case group by reviewing the medical records of each patient according to the inclusion and exclusion criteria. Finally, a control group of patients infected with carbapenem-susceptible *Enterobacterales* (CSE) was consecutively selected in a 1:1 ratio and matched for age, hospitalization period, distribution of services, and sites of infection (Figure 3). The diagnosis of infections was based on the diagnostic criteria for nosocomial infection published by the United States Center for Disease Control (CDC) [36].

Exclusion criteria:-Patients with community-acquired infections (infection diagnosis < 48 h).-Patients with positive culture by colonization or contaminated sample (asymptomatic and negatives in other additional testing).

Inclusion criteria:-Patients with hospital-acquired infections.-This study focused on the primary infection or first episode of CP-CRE-acquired infection during hospitalization if co-infections or recurrent infections occurred.

### 4.4. Data Collection

Demographic data, comorbidities (according to the Charlson index), date of admission and discharge, date of infection, risk factors present before diagnosis of infection (prolonged derivation, deep venous catheterization, urinary catheters or nephrostomy, nasogastric tube, mechanical ventilation, surgical operation, dialysis, previous admission within six months, transfer from other health centers, steroid use, previous antibiotic therapy), subsequent antibiotic treatment of this infection, and clinical outcome were recorded.

### 4.5. Statistical Analysis

Data were described using mean ± SD (quantitative variables) and percentages (qualitative variable). Comparison between groups for metric variables was performed using the Student’s *t*-test while for non-metric variables, a chi-square test or Fisher exact probability test was used as the criterion for determining the statistical significance of differences (*p* ≤ 0.05). For the identification of risk factors for CP-CRE infection, binary logistic regression analysis was applied for multivariate comparison. In multivariate analysis, variables with *p*-values less than 0.15 from univariate analysis were included in the multivariate logistic regression to calculate the odds ratio (OR) and 95% confidence interval. Survival curves (Kaplan–Meier) and Cox regression were used for the mortality analysis. The prevalence ratio (PR) was used in the evaluation of treatment options for CRE infections; however, we added 0.5 to all frequencies when none were available. The analyses were performed with SPSS 22.0 and EPIDAT 3.1.

## 5. Conclusions

The emergence and dissemination of NDM-type carbapenemase-producing *Enterobacterales* associated with HAIs is a great challenge for Cuban public health. From this uni-center, observational study of cases and controls, it is reported that the use of steroids, previous use (more than seven days) of two or more antibiotics, and previous exposure (more than seven days) to cephalosporins (third or fourth generation) or carbapenem are independent factors for the development of CP-CRE infection. Infection with CP-CRE presented a higher mortality compared to that with CSE, especially in terms of bloodstream infections and pneumonia. Although treatment for infection with NDM-producing *Enterobacterales* is still limited, combination therapy remains the currently preferred treatment option, and different combination regimens may have advantages in different types of infections. In bloodstream infections, colistin-based regimens showed a significant advantage while in pneumonia caused by CP-CRE, regimens containing tigecycline showed some advantages.

## Figures and Tables

**Figure 1 antibiotics-11-00942-f001:**
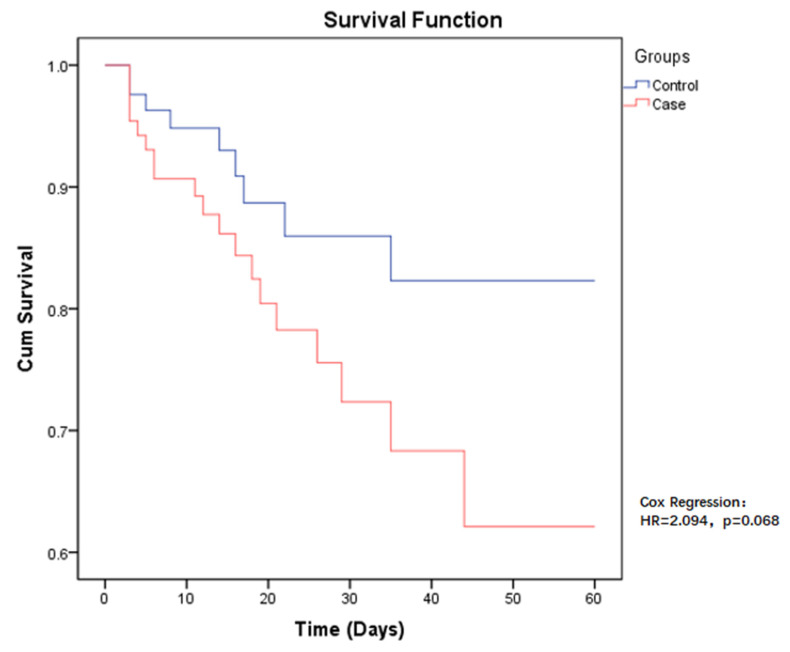
Kaplan–Meier estimates (and 95% confidence limits) of the survival probability of all-cause mortality in CP-CRE (case) and CSE (control) infections, 60-day follow up.

**Figure 2 antibiotics-11-00942-f002:**
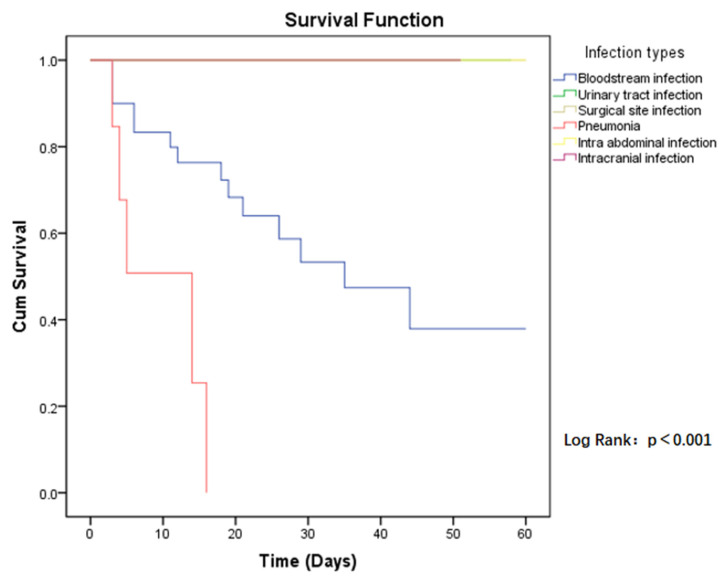
Kaplan–Meier estimates (and 95% confidence limits) of the survival probability of infection types in CP-CRE infections, 60-day follow up.

**Figure 3 antibiotics-11-00942-f003:**
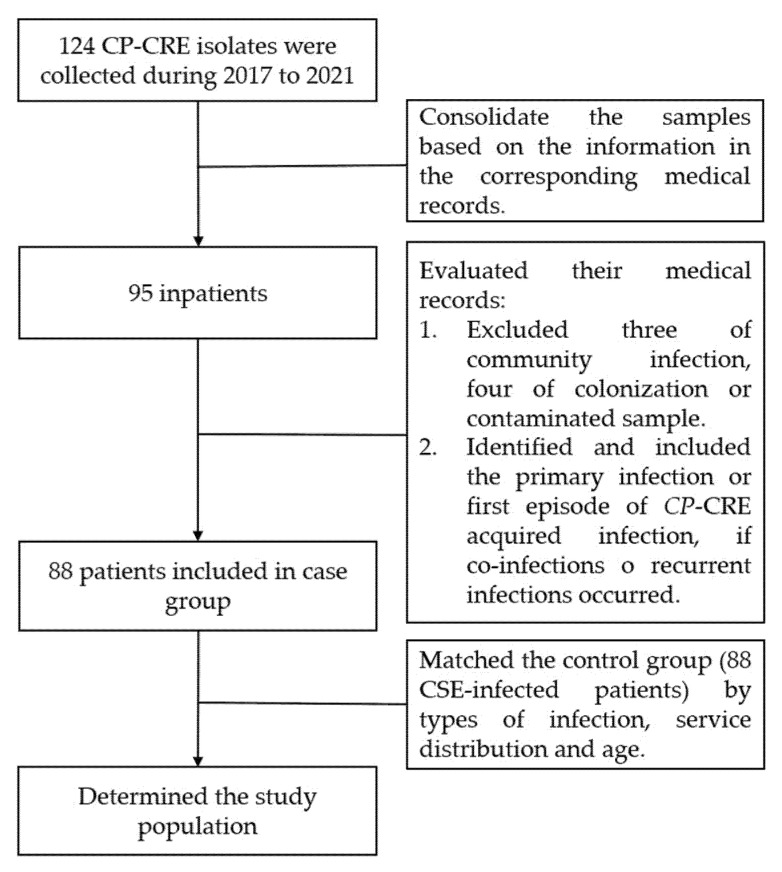
Flow chart of the population identification.

**Table 1 antibiotics-11-00942-t001:** Distribution of carbapenemase types according to methodology and bacterial species (*n* = 124 isolates).

Methods	Carbapenemase Types	Species					
		*K. pneumoniae* (Strains = 88)	*E. cloacae* (Strains = 16)	*E. coli* (Strains = 7)	*K. aerogenes* (Strains = 4)	*S. marcescens* (Strains = 4)	Others ^1^ (Strains = 5)
PCR (55 strains)	NDM	38	5	4	1	1	3
	NDM + KPC	2	0	0	1	0	0
Inmunocromatográfico CORIS (69 strains)	KPC	0	0	0	0	0	0
	NDM	48	11	3	2	3	2
	VIM	0	0	0	0	0	0
	OXA-48	0	0	0	0	0	0

^1^ Others: C. koseri, C. freundii, K. oxytoca, M. morganii.

**Table 2 antibiotics-11-00942-t002:** Antimicrobial resistance rate (%) according to bacterial species (*n* = 124).

Antibiotics ^1^	% Resistance						
	*K. pneumoniae* (Strains = 88)	*E. cloacae* (Strains = 16)	*E. coli* (Strains = 7)	*K. aerogenes* (Strains = 4)	*S. marcescens* (Strains = 4)	Others ^2^ (Strains = 5)	Total (Strains = 124)
SAM	100.0	100.0	100.0	100.0	100.0	100.0	100.0
TZP	100.0	100.0	100.0	100.0	100.0	100.0	100.0
CAZ	100.0	100.0	100.0	100.0	100.0	100.0	100.0
CTX	100.0	100.0	100.0	100.0	100.0	100.0	100.0
FEP	100.0	100.0	100.0	100.0	100.0	100.0	100.0
ATM	97.7	93.8	57.1	75.0	100.0	80.0	93.5
FOS	73.9	75.0	14.3	100.0	75.0	80.0	71.8
CIP	88.6	87.5	57.1	100.0	75.0	100.0	87.1
GEN	92.0	100.0	71.4	100.0	100.0	100.0	92.8
AMK	93.2	93.8	57.1	100.0	100.0	100.0	91.9
SXT	95.5	100.0	57.1	100.0	75.0	100.0	93.5
MRP	100.0	100.0	100.0	100.0	100.0	100.0	100.0
IMI	100.0	100.0	100.0	100.0	100.0	100.0	100.0
CST ^3^	26.1	31.3	0.0	25.0	-	25.0	25.2
TGC	83.0	81.3	42.9	100.0	100.0	60.0	80.6

^1.^ Abbreviation: SAM, ampicillin-sulbactam; TZP, piperacillin-tazobactam; CAZ, ceftazidime; CTX, cefotaxime; FEP, cefepime; ATM, aztreonam; GEN, gentamicin; AMK, amikacin; SXT, trimethoprim-sulfamethoxazole; CIP, ciprofloxacin; FOS, fosfomycin; MEM, meropenem; IPM, imipenem; CST, colistin; TGC, tigecycline. ^2.^ Others: *C. koseri, C. freundii, K. oxytoca, M. morganii.* ^3.^ Resistance rates are not shown for *S. marcescens* and *M. morganii* (intrinsically resistant to CST).

**Table 3 antibiotics-11-00942-t003:** Baseline demographic and clinical data of the population.

Variables	Cases (*n* = 88)	Controls (*n* = 88)	*p*-Value
Mean Age ± SD (Range)	55.5 ± 14.8 (20–87)	53.2 ± 15.5 (22–86)	0.31
**Services**			
Critical Care Unit (ICU y CCU)	25 (28.4%)	23 (26.1%)	0.74
Urology/Lithotripsy	22 (25.0%)	22 (25.0%)	1
General Surgery	7 (8.0%)	6 (6.8%)	0.77
Hematology	4 (4.5%)	4 (4.5%)	1
Nephrology	4 (4.5%)	4 (4.5%)	1
Internal medicine	6 (6.8%)	5 (5.7%)	0.76
Transplantation	4 (4.5%)	6 (6.8%)	0.51
Neurology/neurosurgery	4 (4.5%)	4 (4.5%)	1
Miscellaneous	12 (13.6%)	14 (15.9%)	0.67
**Infection Sites**			
Bloodstream infection	30 (34.1%)	30 (34.1%)	1
Urinary tract infection	30 (34.1%)	31 (35.2%)	0.87
Surgical site infection	18 (20.5%)	18 (20.5%)	1
Pneumonia	7 (8.0%)	6 (6.8%)	0.77
Intra-abdominal infection	2 (2.3%)	2 (2.3%)	1
Intracranial infection	1 (1.1%)	1 (1.1%)	1

**Table 4 antibiotics-11-00942-t004:** Risk factors associated with CRE and CSE infections.

Factors	Cases (*n* = 88)	Controls (*n* = 88)	Univariate Analysis		Multivariate Analysis	
			OR (IC 95%)	*p*-Valor	OR (IC 95%)	*p*-Valor
**Clinical Characteristics**						
Mean Charlson Index score ± SD	3.1 ± 1.9	3.1 ± 2.2	1.01 (0.88–1.17)	0.86	-	-
Average length of hospitalization before infection	27.1 ± 20.0	18.1 ± 14.5	1.03 (1.01–1.05)	<0.01	1.00 (0.97–1.03)	0.95
Previously admitted within the last 6 months	36 (40.9%)	28(31.8%)	1.48 (0.80–2.75)	0.21	-	-
Prolonged derivation (brain, thoracic, abdominal) ^1^	20 (22.7%)	9 (10.2%)	2.58 (1.10–6.05)	0.03	0.93 (0.30–2.95)	0.91
Deep venous catheterization	43 (48.9%)	31 (35.2%)	1.76 (0.96–3.22)	0.07	0.52 (0.19–1.43)	0.21
Urinary catheters (>48 h)	66 (75.0%)	57 (64.8%)	1.63 (0.85–3.13)	0.14	1.52 (0.63–3.68)	0.36
Nasogastric tube	36 (40.9%)	20 (22.7%)	2.35 (1.22–4.53)	0.01	1.45 (0.37–5.76)	0.60
Mechanical ventilation	26 (29.5%)	14 (15.9%)	2.22 (1.07–4.61)	0.03	0.52 (0.12–2.23)	0.34
Surgery	71 (80.7%)	74 (84.1%)	0.79 (0.36–1.72)	0.55	-	-
Dialysis	9 (10.2%)	10 (11.4%)	0.89 (0.34–2.31)	0.81	-	-
Transfer from other health centers	15 (17.0%)	5 (5.7%)	3.41 (1.18–9.84)	0.02	2.98 (0.91–9.69)	0.07
Steroid use	39 (44.3%)	16 (18.2%)	3.58 (1.80–7.11)	<0.01	3.22 (1.36–7.66)	<0.01
Prior use of two or more antibiotics	63 (71.6%)	23 (26.1%)	7.12 (3.67–13.83)	<0.01	4.04 (1.40–11.71)	0.01
**Previous Antibiotic Use (with Course ≥ 7 days)**						
β-lactamase inhibitors	20 (22.7%)	14 (15.9%)	1.56 (0.73–3.32)	0.25	-	-
Cephalosporin first or second generation	8 (9.1%)	4 (4.5%)	2.10 (0.61–7.25)	0.24	-	-
Cephalosporin third or fourth generation	53 (60.2%)	26 (29.5%)	3.61 (1.93–6.75)	<0.01	2.40 (1.06–5.44)	0.04
Aminoglycoside	38 (43.2%)	13 (14.8%)	4.39 (2.13–9.05)	<0.01	2.06 (0.74–5.72)	0.16
Quinolone	31 (35.2%)	21 (23.9%)	1.74 (0.90–3.35)	0.1	0.78 (0.28–2.17)	0.63
Carbapenem	27 (30.7%)	4 (4.5%)	9.30 (3.09–27.94)	<0.01	4.77 (1.17–19.35)	0.03
Sulfonamide	11 (12.5%)	6 (6.8%)	1.95 (0.69–5.54)	0.21	-	-

^1^ Prolonged derivation: brain derivation > 5 days; thoracic derivation > 3 days; abdominal derivation > 3 days.

**Table 5 antibiotics-11-00942-t005:** Univariate analysis of treatment options associated with 60-day mortality.

Treatment Options ^1^	Bloodstream Infection				Pneumonia			
	Survival (Case = 16)	Non-Survival (Case = 14)	PR	*p*-Value	Survival (Case = 2)	Non-Survival (Case = 5)	PR	*p*-Value
Monotherapy	3 (18.6%)	5 (35.7%)	1.53 (0.73–3.19)	0.26	0	2 (40.0%)	1.43 (0.61–3.32)	0.48
Colistin-based	11 (68.8%)	4 (28.6%)	0.40 (0.16–0.99)	0.03	0	3 (60.0%)	1.75 (0.68–4.53)	0.29
Tigecycline-based	0	2 (14.3%)	1.93 (1.00–3.72)	0.21	1 (50.0%)	0	0.52 (0.13–1.96)	0.28
Colistin-tigecycline-based	1 (6.25%)	2 (14.3%)	1.50 (0.61–3.71)	0.45	1 (50.0%)	0	0.52 (0.13–1.97)	0.28

^1^ Colistin-based: colistin/aminoglycosides or fluoroquinolone or fosfomycin or carbapenem. Tigecycline-based: tigecycline/aminoglycosides or fosfomycin. Colistin-tigecycline-based: colistin-tigecycline or colistin-tigecycline/fosfomycin.

**Table 6 antibiotics-11-00942-t006:** Oligonucleotides used in this study.

Primer	Sequence (5′–3′)	Gene	Product Size (bp)
KPC-Fm	CGTCTAGTTCTGCTGTCTTG	*bla*KPC	798
KPC-Rm	CTTGTCATCCTTGTTAGGCG		
NDM-F	GGTTTGGCGATCTGGTTTTC	*bla*NDM	621
NDM-R	CGGAATGGCTCATCACGATC		
IMP-F	GGAATAGAGTGGCTTAAYTCTC	*bla*IMP	232
IMP-R	GGTTTAAYAAAACAACCACC		
VIM-F	GATGGTGTTTGGTCGCATA	*bla*VIM	390
VIM-R	CGAATGCGCAGCACCAG		
SPM-F	AAAATCTGGGTACGCAAACG	*bla*SPM	271
SPM-R	ACATTATCCGCTGGAACAGG		
GIM-F	TCGACACACCTTGGTCTGAA	*bla*GIM	477
GIM-R	AACTTCCAACTTTGCCATGC		
SIM-F	TACAAGGGATTCGGCATCG	*bla*SIM	570
SIM-R	TAATGGCCTGTTCCCATGTG		
OXA-F	GCGTGGTTAAGGATGAACAC	*bla*OXA-48	438
OXA-R	CATCAAGTTCAACCCAACCG		

## Data Availability

The datasets generated during and/or analyzed during the current study are available from the corresponding author on reasonable request.

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
