# Peer review of "A Retrospective Study of Risk Factors, Mortality, and Treatment Outcomes for Infections with Carbapenemase-Producing Enterobacterales in a Tertiary Hospital in Havana, Cuba"

_antibiotics, 2022, doi:10.3390/antibiotics11070942_

Round 1

Reviewer 1 Report

This is a descriptive, observational, retrospec- 16 tive case-control study of patients infected with both carbapenemase-producing and carbapenem-susceptible Enterobacterales. 124 CRE isolates were checked, mostly found to be NDM carriers and a 3 KPC. Risk factors for mortality were assessed. The study shows that the infection of CP-CRE presented a higher mortality compared to CSE, especially in bloodstream infection and pneumonia. Colistin-based treatments were better for bloodstream infections.

Major comments:

Title: the title seems to have little relevance to the actual study discussed, as the study centers around the mortality risks and desired treatment of NDM infections than the actual NDM carrying isolates. A title addressing the conclusions would be more fitting.

Lines 67-70: Why did you choose to use two separate methods? Was there a logic to this or was it accidental? Did isolates that were found negative by one method undergo an additional assessment using the second method?

Line 72: The fact that colistin resistance is widespread in many countries, while it is rare in Latin America should be at least mentioned.

Table 2: I do not think a table giving a range as wide as 0.016-256 can in any way help our understanding of the data. Also the term 'non-susceptibility' is confusing. I would remove this table and find another way to sum up the results that is less confusing.

Line 127 – close to the limit is still not significant. This sentence could be better phrased "…the two groups was not significantly different (HR=2.094, p=0.068) (Figure 1).

Lines 176-219: please edit these paragraphs, they are confusing and not well written. For example, why isn't line 210 a new paragraph? It is a new topic.

Minor comments:

Line 173-174: "Another similar study, except that carbapenem use was consistent with this study, length of hospitalization and invasive procedures were also independent risk factors for the infection of CRE.[12]" - Please edit it to be clearer, it is hard to decipher.

Line 185: " …possibly due to the traveling from endemic areas and after local dissemination" – this sentence too requires editing.

Lines 248-249: " Currently, studies related to treatment options for CP-CRE usually focus on 248 KPC or OXA-48 carbapenemase-producing… " is there something missing in this sentence? It seems to jump between two distinct topics in one go.

Line 261: "Limitations." – Something is wrong in the wording – did you mean to write 'limitations; "? it seems incongruous.

Author Response

First, we thank the reviewers for their valuable suggestions, which could lead to a more complete presentation of the study results. We have made item-by-item changes based on your suggestions.

Based on your suggestions, we have applied for English editing services to correct the language expressions in the all text.

Major comments:

1. Title: the title seems to have little relevance to the actual study discussed, as the study centers around the mortality risks and desired treatment of NDM infections than the actual NDM carrying isolates. A title addressing the conclusions would be more fitting.

R: We have modified the title as you suggested and the title is as follows: “A retrospective study on risk factors, mortality and treatment outcomes for infections with carbapenemase-producing Enterobacterales in a tertiary hospital in Havana, Cuba”

2. Lines 67-70: Why did you choose to use two separate methods? Was there a logic to this or was it accidental? Did isolates that were found negative by one method undergo an additional assessment using the second method?

R: Cuba is a low-resource country,this study was originally designed to detect carbapenemases in all isolates by PCR, but due to the impact of Covid-19 pandemic, the PCR instrument needed to prioritize the detection of Covid-19, so we used the CORIS rapid test to detect some isolates. However, we consider that the results are reliable and accurate, as several studies have shown the very high sensitivity (97.8%-99.2%) and specificity (100%) of this method. We hope you would understand.

Related references are as follows:

  1. Greissl, C., Saleh, A., & Hamprecht, A. (2018). Rapid detection of OXA-48-like, KPC, NDM, and VIM carbapenemases in Enterobacterales by a new multiplex immunochromatographic test. European Journal of Clinical Microbiology & Infectious Diseases.doi:10.1007/s10096-018-3432-2
  2. Song W, Park MJ, Jeong S, Shin DH, Kim JS, Kim HS, Kim HS, Lee N, Hong JS, Jeong SH. Rapid Identification of OXA-48-like, KPC, NDM, and VIM Carbapenemase-Producing Enterobacteriaceae From Culture: Evaluation of the RESIST-4 O.K.N.V. Multiplex Lateral Flow Assay. Ann Lab Med. 2020 May;40(3):259-263. doi: 10.3343/alm.2020.40.3.259.

3. Line 72: The fact that colistin resistance is widespread in many countries, while it is rare in Latin America should be at least mentioned.

R: This view has been added to the discussion section in line 208-214: “However, it should be noted that, although the rate of colistin resistance in this study was relatively low, it still far exceeded the overall level of colistin resistance in Latin America and even reached that seen in some European countries (Spain, Italy, Greece, etc.), with relatively high levels of colistin resistance. A review study (2022) shows that the incidence of colistin-resistant human clinical Enterobacterales increased between 2014 and 2019 from 2.7 in Latin America to 4.3%, while the colistin resistance rates in these European countries have stayed between 20% and 40%.”

4. Table 2: I do not think a table giving a range as wide as 0.016-256 can in any way help our understanding of the data. Also the term 'non-susceptibility' is confusing. I would remove this table and find another way to sum up the results that is less confusing.

R: Table 2 has been modified, the new table is “Antimicrobial resistance rate (%) according to bacterial species (n = 124)”.

5. Line 127 – close to the limit is still not significant. This sentence could be better phrased "…the two groups was not significantly different (HR=2.094, p=0.068) (Figure 1).

R: We have modified in line 128-129: “The risk of mortality between the two groups, the p-value was close to the limit and was still not significant (HR=2.094, p=0.068)”.

6. Lines 176-219: please edit these paragraphs, they are confusing and not well written. For example, why isn't line 210 a new paragraph? It is a new topic.

R: We have revised all the language expressions in this section and adjusted the structure of the paragraphs. line177-225.

Minor comments:

7. Line 173-174: "Another similar study, except that carbapenem use was consistent with this study, length of hospitalization and invasive procedures were also independent risk factors for the infection of CRE.[12]" - Please edit it to be clearer, it is hard to decipher.

R: We have modified in line 171-173: “However, another case–control study of CRE and CSE found that carbapenem use, length of hospitalization, and invasive procedures were also independent risk factors.”

8. Line 185: " …possibly due to the traveling from endemic areas and after local dissemination" – this sentence too requires editing.

R: Worldwide, NDM producers are most common in Asia and Europe, while KPC is still predominant in Latin America. However, Cuba is also heavily affected by NDM-producing Enterobacterales, also because Cuba is an important tourist country in Latin America, it is assumed that the NDM epidemic in Cuba may be due to travel from endemic areas and result in silent local dissemination.

9. Lines 248-249: " Currently, studies related to treatment options for CP-CRE usually focus on 248 KPC or OXA-48 carbapenemase-producing… " is there something missing in this sentence? It seems to jump between two distinct topics in one go.

R: This paragraph has been modified in line 254-261: “For MBL-producing Enterobacterales—in particular, those that co-produce serine-type β-lactamases—treatment is more challenging. Currently, studies related to treatment options for CP-CRE usually focus on KPC or OXA-48 carbapenemase-producing Enterobacterales, and it has been found that, in addition to combinations based on tigecycline, polymyxin, or carbapenems which present a synergistic effect, newer drugs such as ceftazidime-avibactam, meropenem-vaborbactam, and plazomycin, possess greater advantages against carbapenemase KPC- or OXA-48-producing Enterobacterales.”

10. Line 261: "Limitations." – Something is wrong in the wording – did you mean to write 'limitations; "? it seems incongruous.

Yes , we describe some of the limitations of this study.

Reviewer 2 Report

This paper looks at the epidemiological characteristics of infection with carbapenemase producing Enterobacterales in a tertiary hospital in Havana in Cuba.

The paper is overall well written. I am happy to recommend subject to dealing with some minor comments and suggestions.

Were the types of Enterobacterales identified (and their antibiotic resistance patterns other then for carbapenem of course) similar for the carbapenem susceptible isolates?

Please add the Primers used and the PCR condition to the Materials and Methods.

Author Response

We thank the reviewer for your valuable suggestion, which could make the results of the study more complete in presentation.

1. Were the types of Enterobacterales identified (and their antibiotic resistance patterns other then for carbapenem of course) similar for the carbapenem susceptible isolates?

R: First of all, we need to explain to you that species of CSE are essentially the same as CRE. In addition, according to the national bacterial resistance surveillance program, all CRE strains detected in the hospital need to be sent to National Reference Laboratory for Healthcare-Associated Infections for verification and further testing. All CRE strains in this study were verified by our laboratory, but for CSE the results were derived from the hospital laboratory results and were not secondarily verified by our laboratory, so we did not present the results in the study. However, according to the results of the hospital laboratory, it can be observed that CSE is much more sensitive than CRE strains for b-lactam, fluoroquinolone, and aminoglycoside. Finally, we hope you would understand.

2: Please add the Primers used and the PCR condition to the Materials and Methods.

R: We have added the Primers used (table 6) and the PCR condition in line 306-316: “Briefly, eight pairs of primers were designed to amplify internal fragments with sizes from 232 to 798 bp (Table 6),the thermal cycling settings were 30 cycles at 95℃ for 30 s, 55℃ for 1 min, and 72℃ for 30 s, as well as one cycle at 72℃ for 5 min. A positive control was used for the amplification of each gene. The PCR products were separated by electrophoresis on 1.5% agarose gel (AppliChem, Germany) and visualized using UV transillumination (Uvitec, United Kingdom).”
